# Mapping epigenetic divergence in the massive radiation of Lake Malawi cichlid fishes

Grégoire Vernaz [1,2,3✉], Milan Malinsky [3,7], Hannes Svardal[3,8,9], Mingliu Du[1,2,3], Alexandra M. Tyers[4,10], M. Emília Santos [5], Richard Durbin [2,3], Martin J. Genner [6], George F. Turner [4] & Eric A. Miska [1,2,3✉]

Epigenetic variation modulates gene expression and can be heritable. However, knowledge of the contribution of epigenetic divergence to adaptive diversification in nature remains limited. The massive evolutionary radiation of Lake Malawi cichlid fishes displaying extensive phenotypic diversity despite extremely low sequence divergence is an excellent system to study the epigenomic contribution to adaptation. Here, we present a comparative genome-wide methylome and transcriptome study, focussing on liver and muscle tissues in phenotypically divergent cichlid species. In both tissues we find substantial methylome divergence among species. Differentially methylated regions (DMR), enriched in evolutionary young transposons, are associated with transcription changes of ecologically-relevant genes related to energy expenditure and lipid metabolism, pointing to a link between dietary ecology and methylome divergence. Unexpectedly, half of all species-specific DMRs are shared across tissues and are enriched in developmental genes, likely reflecting distinct epigenetic developmental programmes. Our study reveals substantial methylome divergence in closely-related cichlid fishes and represents a resource to study the role of epigenetics in species diversification.

[1] Wellcome/CRUK Gurdon Institute, University of Cambridge, Cambridge, UK. [2] Department of Genetics, University of Cambridge, Cambridge, UK. [3] Wellcome Sanger Institute, Cambridge, UK. [4] School of Natural Sciences, Sciences, Bangor University, Bangor, UK. [5] Department of Zoology, University of Cambridge, Cambridge, UK. [6] School of Biological Sciences, University of Bristol, Bristol, UK. [7] Present address: Institute of Ecology and Evolution, University of Bern, Bern, Switzerland. [8] Present address: Department of Biology, University of Antwerp, Antwerp, Belgium. [9] Present address: Naturalis Biodiversity Center, Leiden, The Netherlands. [10] Present address: Max Planck Institute for Biology of Ageing, Cologne, Germany. ✉email: gv268@cam.ac.uk; eam29@cam.ac.uk

Trait inheritance and phenotypic diversification are primarily explained by the transmission of genetic information encoded in the DNA sequence. In addition, a variety of epigenetic processes have recently been reported to mediate heritable transmission of phenotypes in animals and plants[1–7]. However, the current understanding of the evolutionary significance of epigenetic processes, and of their roles in organismal diversification, is in its infancy.

DNA methylation, or the covalent addition of a methyl group onto the 5th carbon of cytosine (mC) in DNA, is a reversible epigenetic mark present across multiple kingdoms[8–10], can be heritable, and has been linked to transmission of acquired phenotypes in plants and animals[2,5,6,11–13]. The importance of this mechanism is underlined by the fact that proteins involved in the deposition of mC ('writers', DNA methyltransferases [DNMTs]), in mC maintenance during cell division, and in the removal of mC ('erasers', ten-eleven translocation methylcytosine dioxygenases [TETs]), are mostly essential and show high degrees of conservation across vertebrates species[14–17]. In addition, some ancestral functions of methylated cytosines are highly conserved, such as in the transcriptional silencing of exogenous genomic elements (transposons)[18,19]. In vertebrates, DNA methylation functions have evolved to play an important role in the orchestration of cell differentiation during normal embryogenesis/development through complex interactions with histone post-translational modifications (DNA accessibility) and mC-sensitive readers (such as transcription factors)[19–25], in particular at cis-regulatory regions (i.e., promoters, enhancers). Early-life establishment of stable DNA methylation patterns can thus affect transcriptional activity in the embryo and persist into fully differentiated cells[26]. DNA methylation variation has also been postulated to have evolved in the context of natural selection by promoting phenotypic plasticity and thus possibly facilitating adaptation, speciation, and adaptive radiation[2,4,12,27].

Studies in plants have revealed how covarying environmental factors and DNA methylation variation underlie stable and heritable transcriptional changes in adaptive traits[2,6,11–13,28]. Some initial evidence is also present in vertebrates[2,5,29–31]. In the cavefish, for example, an early developmental process—eye degeneration—has been shown to be mediated by DNA methylation, suggesting mC variation as an evolutionary factor generating adaptive phenotypic plasticity during development and evolution[29,32]. However, whether correlations between environmental variation and DNA methylation patterns promote phenotypic diversification more widely among natural vertebrate populations remains unknown.

In this study, we sought to quantify, map and characterise natural divergence in DNA methylation in the context of the Lake Malawi haplochromine cichlid adaptive radiation, one of the most spectacular examples of rapid vertebrate phenotypic diversification[33]. In total, the radiation comprises over 800 endemic species[34], that are estimated to have evolved from common ancestry approximately 800,000 years ago[35]. Species within the radiation can be grouped into seven distinct ecomorphological groups based on their ecology, morphology, and genetic differences: (1) shallow benthic, (2) deep benthic, (3) deep pelagic zooplanktivorous/piscivorous *Diplotaxodon*, (4) the rock-dwelling 'mbuna', (5) zooplanktivorous 'utaka', (6) *Astatotilapia calliptera* specialised for shallow weedy habitats (also found in surrounding rivers and lakes), and (7) the midwater pelagic piscivores *Rhamphochromis*[36,37]. Recent large-scale genetic studies have revealed that the Lake Malawi cichlid flock is characterised by an overall very low genetic divergence among species (0.1–0.25%), combined with a low mutation rate, a high rate of hybridisation and extensive incomplete lineage sorting (shared retention of ancestral genetic variation across species)[34,36,38,39].

Multiple molecular mechanisms may be at work to enable such an explosive phenotypic diversification. Therefore, investigating the epigenetic mechanisms in Lake Malawi cichlids represents a remarkable opportunity to expand our comprehension of the processes underlying phenotypic diversification and adaptation.

Here we describe, quantify, and assess the divergence in liver methylomes in six cichlid species spanning five of the seven ecomorphological groups of the Lake Malawi haplochromine radiation by generating high-coverage whole-genome liver bisulfite sequencing (WGBS). We find that Lake Malawi haplochromine cichlids exhibit substantial methylome divergence, despite conserved underlying DNA sequences, and are enriched in evolutionary young transposable elements. Next, we generated whole liver transcriptome sequencing (RNAseq) in four of the six species and showed that differential transcriptional activity is significantly associated with between-species methylome divergence, most prominently in genes involved in key hepatic metabolic functions. Finally, by generating WGBS from muscle tissues in three cichlid species, we show that half of methylome divergence between species is tissue-unspecific and pertains to embryonic and developmental processes, possibly contributing to the early establishment of phenotypic diversity. This represents a comparative analysis of natural methylome variation in Lake Malawi cichlids and provides initial evidence for substantial species-specific epigenetic divergence in cis-regulatory regions of ecologically-relevant genes. Our study represents a resource that lays the groundwork for future epigenomic research in the context of phenotypic diversification and adaptation.

## Results

**The methylomes of Lake Malawi cichlids feature conserved vertebrate characteristics.** To characterise the methylome variation and assess possible functional relationships in natural populations of Lake Malawi cichlids, we performed high-coverage whole-genome bisulfite sequencing of methylomes (WGBS) from liver tissues of six different cichlid species. Muscle methylome (WGBS) data for three of the six species were also generated to assess the extent to which methylome divergence was tissue-specific. Moreover, to examine the correlation between transcriptome and methylome divergences, total transcriptomes (RNAseq) from both liver and muscle tissues of four species were generated. Only wild-caught male specimens (2−3 biological replicates for each tissue and each species) were used for all sequencing datasets (Fig. 1a–c, Supplementary Fig. 1, Supplementary Data 1, and Supplementary Table 1). The species selected were: *Rhamphochromis longiceps* (RL), a pelagic piscivore (Rhamphochromis group); *Diplotaxodon limnothrissa* (DL), a deep-water pelagic carnivore (Diplotaxodon group); *Maylandia zebra* (MZ) and *Petrotilapia genalutea* (PG), two rock-dwelling algae eaters (Mbuna group); *Aulonocara stuartgranti* (AS), a benthic invertebrate-eating sand/rock-dweller that is genetically part of the deep-benthic group; *Astatotilapia calliptera* (AC), a species of rivers and lake margins[40] (Fig. 1b).

On average, 285.51 ± 55.6 million paired-end reads (see Supplementary Data 1) for liver and muscle methylomes were generated with WGBS, yielding ~10−15x per-sample coverage at CG dinucleotide sites (Supplementary Fig. 2a–d; see "Methods" and Supplementary Notes). To account for species-specific genotype and avoid methylation biases due to species-specific single nucleotide polymorphism (SNP), WGBS reads were mapped to SNP-corrected versions of the *Maylandia zebra* reference genome (UMD2a; see Methods). Mapping rates were not significantly different among all WGBS samples (Dunn's test with Bonferroni correction, $p > 0.05$; Supplementary Fig. 2e), reflecting the high level of conservation at the DNA sequence

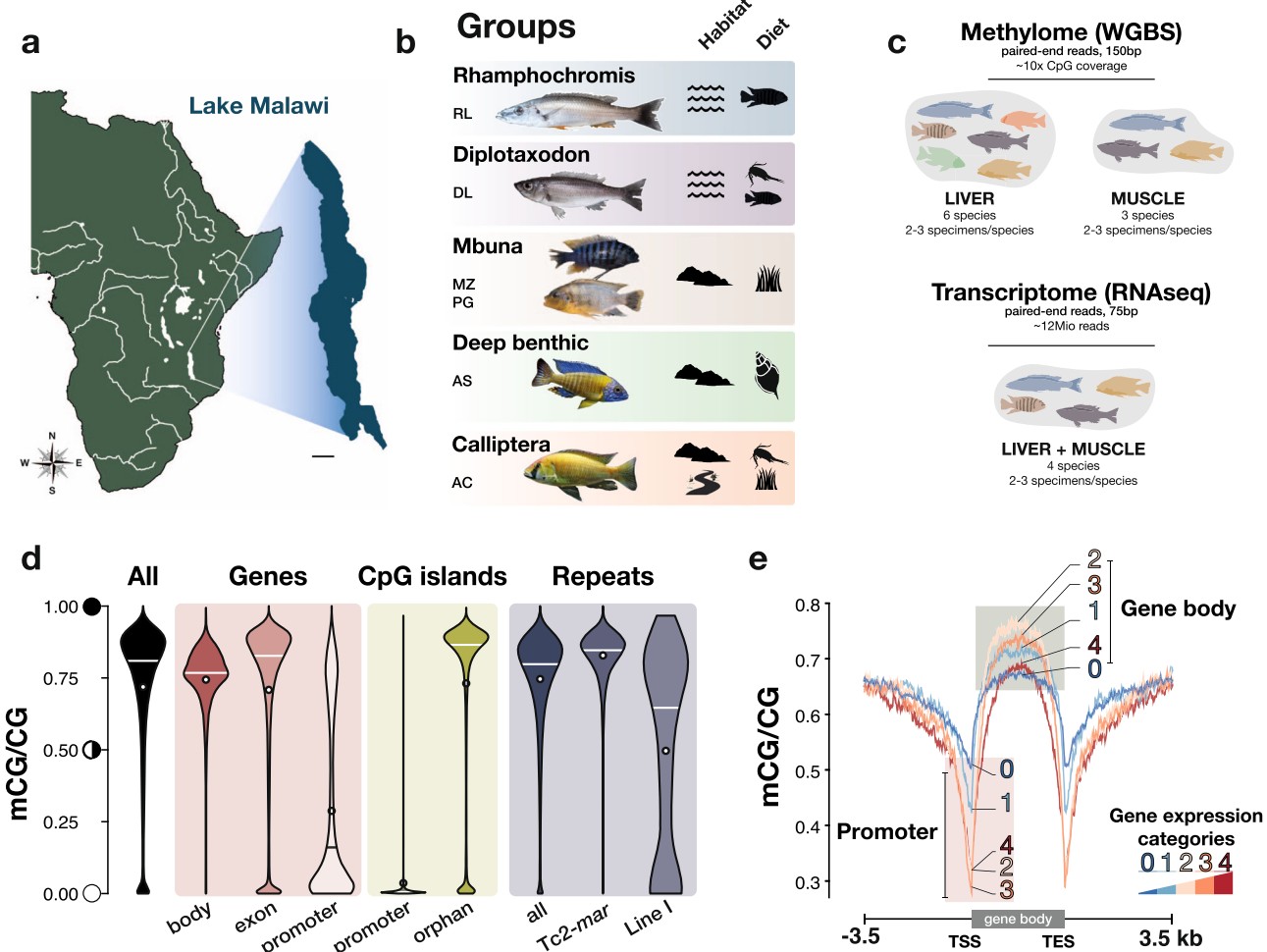

**Fig. 1 The methylome of Lake Malawi cichlids. a** Map of Africa (main river systems are highlighted in white) and magnification of Lake Malawi (scale bar: 40 km). **b** Photographs (not to scale) of the six Lake Malawi cichlid species part of this study spanning five of the seven described eco-morphological groups. The symbols represent the different habitats (pelagic/benthic [wave symbol], rock/sand-dwelling/littoral [rock symbol] and adjacent rivers part of Lake Malawi catchment), and the type of diet (fish, fish/zooplankton, algae, invertebrates) for each group. The species representing each group are indicated by their initials (see below). **c** Diagram summarising the sampling and sequencing strategies for liver and muscle methylome (whole-genome bisulfite sequencing, WGBS) and whole transcriptome (RNAseq) datasets. See "Methods", Supplementary Fig. 1 and Supplementary Table 1. **d** Violin plots showing the distribution of liver DNA methylation levels in CG sequence context (averaged mCG/CG levels over 50 bp-long bins genome-wide) in different genomic regions: overall, gene bodies, exons, promoter regions (TSS ± 500 bp), CpG-islands in promoters and outside (orphan) and in repeat/transposon regions. mC levels for two different repeat classes are given: DNA transposon superfamily Tc2-Mariner ($n = 5,378$) and LINE I ($n = 407$). **e** Average liver mCG profiles across genes differ depending on their transcriptional activity in liver: from non-expressed (0) to genes showing low (1), intermediate (2), high (3) and highest (4) expression levels ("Methods"). Results shown in (**d, e**) are for Mbuna MZ (liver, $n = 3$) and are representative of the results for all other species, and are based on average mC/C in 50 bp non-overlapping windows. *RL, Rhamphochromis longiceps; DL, Diplotaxodon limnothrissa; MZ, Maylandia zebra; PG, Petrotilapia genalutea; AS, Aulonocara stuartgrandti; AC, Astatotilapia calliptera.* Credits—Fish photographs: Hannes Svardal and M. Emília Santos. Geographical map modified from www.d-maps.com/.

level across the Malawi radiation (Supplementary Fig. 3). In parallel, liver and muscle transcriptomes were generated for four species using the same specimens as used for WGBS, yielding on average 11.9 ± 0.7 million paired-end reads (mean ± sd; Fig. 1c, Supplementary Data 1 and "Methods").

We first characterised global features of the methylome of Lake Malawi cichlids. The genome of Lake Malawi cichlid was found to have copies of DNA methyltransferases (DNMTs) and ten-eleven translocation methylcytosine dioxygenases (TETs), the 'readers' and 'erasers' of DNA methylation respectively (Supplementary Fig. 4a−c). Like that of mammals and other teleost fish, the genomes of Lake Malawi cichlids have high levels of DNA methylation genome-wide in the CG dinucleotide sequence context, consistently across all samples in both tissues analysed (Fig. 1d and Supplementary Fig. 2a−c). Gene bodies generally

show higher methylation levels than the genome-wide average, while the majority of promoter regions are unmethylated (Fig. 1d). CpG islands (CGIs; i.e., CpG-rich regions—abundant in Lake Malawi cichlid genomes; Supplementary Fig. 5a−i, Supplementary Notes and Methods) are almost entirely devoid of methylation in promoters, while 'orphan' CGIs, residing outside promoters, are mostly highly methylated (Fig. 1d and Supplementary Fig. 5f, g). While 70% of mammalian promoters contain CGIs[41], only 15−20% of promoters in Lake Malawi cichlids harbour CGIs (Supplementary Fig. 5d), similar to frog and zebrafish genomes[41]. Notably, orphan CGIs, which may have important *cis*-regulatory functions[42], compose up to 80% of all predicted CGIs in Lake Malawi cichlids (Supplementary Fig. 5e). Furthermore, repetitive regions, as well as transposable elements, are particularly enriched for cytosine methylation, suggesting a

methylation-mediated silencing of their transcription (Fig. 1d, Supplementary Fig. 6a−d), similar to that observed in zebrafish and other animals[8,18]. Interestingly, certain transposon families, such as LINE I and Tc2-Mariner, part of the DNA transposon family—the most abundant TE family predicted in Lake Malawi cichlid genome (Supplementary Fig. 6a, b, Supplementary Notes, and ref. [38])—have recently expanded considerably in the Mbuna genome (Supplementary Fig. 6c and refs. [38,43]). While Tc2-Mar DNA transposons show the highest median methylation levels, LINE I elements have some of the lowest, yet most variable, methylation levels of all transposon families, which correlates with their evolutionary recent expansion in the genome (Fig. 1d, e and Supplementary Fig. 6d, e). Finally, transcriptional activity in liver and muscle tissues of Lake Malawi cichlids was negatively correlated with methylation in promoter regions (Spearman's correlation test, $\rho = -0.40$, $p < 0.002$), while being weakly positively correlated with methylation in gene bodies ($\rho = 0.1$, $p < 0.002$; Fig. 1e and Supplementary Fig. 7a−d and Supplementary Table 2). This is consistent with previous studies highlighting high methylation levels in bodies of active genes in plants and animals, and high levels of methylation at promoters of weakly expressed genes in vertebrates[8,24]. We conclude that the methylomes of Lake Malawi cichlids share many regulatory features, and possibly associated functions, with those of other vertebrates, which renders Lake Malawi cichlids a promising model system in this context.

**Methylome divergence in Lake Malawi cichlids**. To assess the possible role of DNA methylation in phenotypic diversification, we then sought to quantify and characterise the differences in liver and muscle methylomes across the genomes of Lake Malawi haplochromine cichlids. Despite overall very low sequence divergence[36] (Supplementary Fig. 3), Lake Malawi cichlids were found to show substantial methylome divergence across species within each tissue type, while within-species biological replicates always clustered together (Fig. 2a). The species relationships inferred by clustering of the liver methylomes at conserved individual CG dinucleotides reca-pitulate some of the genetic relationship inferred from DNA sequence[36], with one exception—the methylome clusters A. calliptera samples as an outgroup, not a sister group to Mbuna (Fig. 2a and Supplementary Fig. 3a, b). This is consistent with its unique position as a riverine species, while all species are obligate lake dwellers (Fig. 1b).

As DNA methylation variation tends to correlate over genomic regions consisting of several neighbouring CG sites, we defined and sought to characterise differentially methylated regions (DMRs) among Lake Malawi cichlid species (≥50 bp-long, ≥4 CG dinucleotide, and ≥25% methylation difference across any pair of species, $p < 0.05$; see Methods). In total, 13,331 between-species DMRs were found among the liver methylomes of the six cichlid species (Supplementary Fig. 8a). We then compared the three species for which liver and muscle WGBS data were available and found 5,875 and 4,290 DMRs among the liver and muscle methylomes, respectively. By contrast, 27,165 within-species DMRs were found in the between-tissue comparisons (Supplementary Fig. 8b). Overall, DMRs in Lake Malawi cichlids were predicted to be as long as 5,000 bp (95% CI of median size: 282−298 bp; Supplementary Fig. 8c). While the methylation differences between liver and muscle were the most prominent at single CG dinucleotide resolution (Fig. 2a) and resulted in the highest number of DMRs, we found DMRs to be slightly larger and methylation differences within them substantially stronger among species than between tissues (Dunn's test, $p < 2.2 \times 10^{-16}$; Supplementary Fig. 8c, d).

Next, we characterised the genomic features enriched for between-species methylome divergence in the three cichlid species for which both muscle and liver WGBS data were available (i.e., RL, PG, DL; Fig. 1c). In the liver, promoter regions and orphan CGIs have 3.0- and 3.6-fold enrichment respectively for between-species liver DMRs over random expectation ($\chi^2$ test, $p < 0.0001$; Fig. 2b)—between-species muscle DMRs show similar patterns as well ($p = 0.99$, compared to liver O/E ratios). Methylome variation at promoter regions has been shown to affect transcription activity via a number of mechanisms (e.g., transcription factor binding affinity, chromatin accessibility)[21,44] and, in this way, may participate in phenotypic adaptive diversification in Lake Malawi cichlids. In particular, genes with DMRs in their promoter regions show enrichment for enzymes involved in hepatic metabolic functions (Fig. 2c). Furthermore, the high enrichment of DMRs in intergenic orphan CGIs (Fig. 2b), accounting for $n = 691$ (11.94%) of total liver DMRs, suggests that intergenic CGIs may have DNA methylation-mediated regulatory functions.

The majority of between-species liver DMRs (65.0%, $n = 3,764$) are within TE regions (TE-DMRs; Supplementary Fig. 8a, b, e), approximately two-thirds of which are located in unannotated intergenic regions (Fig. 2d). However, a small fraction of TE-DMRs are located in gene promoters (12% of all TE-DMRs) and are significantly enriched in genes associated with metabolic pathways (Fig. 2d and Supplementary Fig. 8f). While there is only a 1.1-fold enrichment of DMRs globally across all TEs (Fig. 2b), some TE families are particularly enriched for DMRs, most notably the DNA transposons hAT (hAT6, 10.5-fold), LINE/l (>3.7-fold) and the retrotransposons SINE/Alu (>3.5-fold). On the other hand, the degree of methylation in a number of other TE families shows unexpected conservation among species, with substantial DMR depletion (e.g., LINE/R2-Hero, DNA/Maverick; Fig. 2e). Overall, we observe a pattern whereby between-species methylome differences are significantly localised in younger transposon sequences (Dunn's test, $p = 2.2 \times 10^{-16}$; Fig. 2f). Differential methylation in TE sequences may affect their transcription and transposition activities, possibly altering or establishing new transcriptional activity networks via cis-regulatory functions[45–47]. Indeed, the movement of transposable elements has recently been shown to contribute to phenotypic diversification in Lake Malawi cichlids[48].

In contrast to the between-species liver DMRs, within-species DMRs based on comparison of liver against muscle methylomes show much less variation in enrichment across genomic features. Only gene bodies show weak enrichment for methylome variation (Fig. 2b). Moreover, both CGI classes, as well as repetitive and intergenic regions show considerable tissue-DMR depletion, suggesting a smaller DNA methylation-related contribution of these elements to tissue differentiation (Fig. 2b and Supplementary Fig. 8e).

**Methylome divergence is associated with transcriptional changes in the livers**. We hypothesised that adaptation to different diets in Lake Malawi cichlids could be associated with distinct hepatic functions, manifesting as differences in transcriptional patterns which, in turn, could be influenced by divergent methylation patterns. To investigate this, we first performed differential gene expression analysis. In total, 3,437 genes were found to be differentially expressed between livers of the four Lake Malawi cichlid species investigated (RL, DL, MZ, and PG; Wald test, false discovery rate adjusted two sided $p$-value using Benjamini−Hochberg [FDR] < 0.01; Fig. 3a and Supplementary Fig. 9a−c; see "Methods"). As with methylome variation, transcriptome variation clustered individuals by species

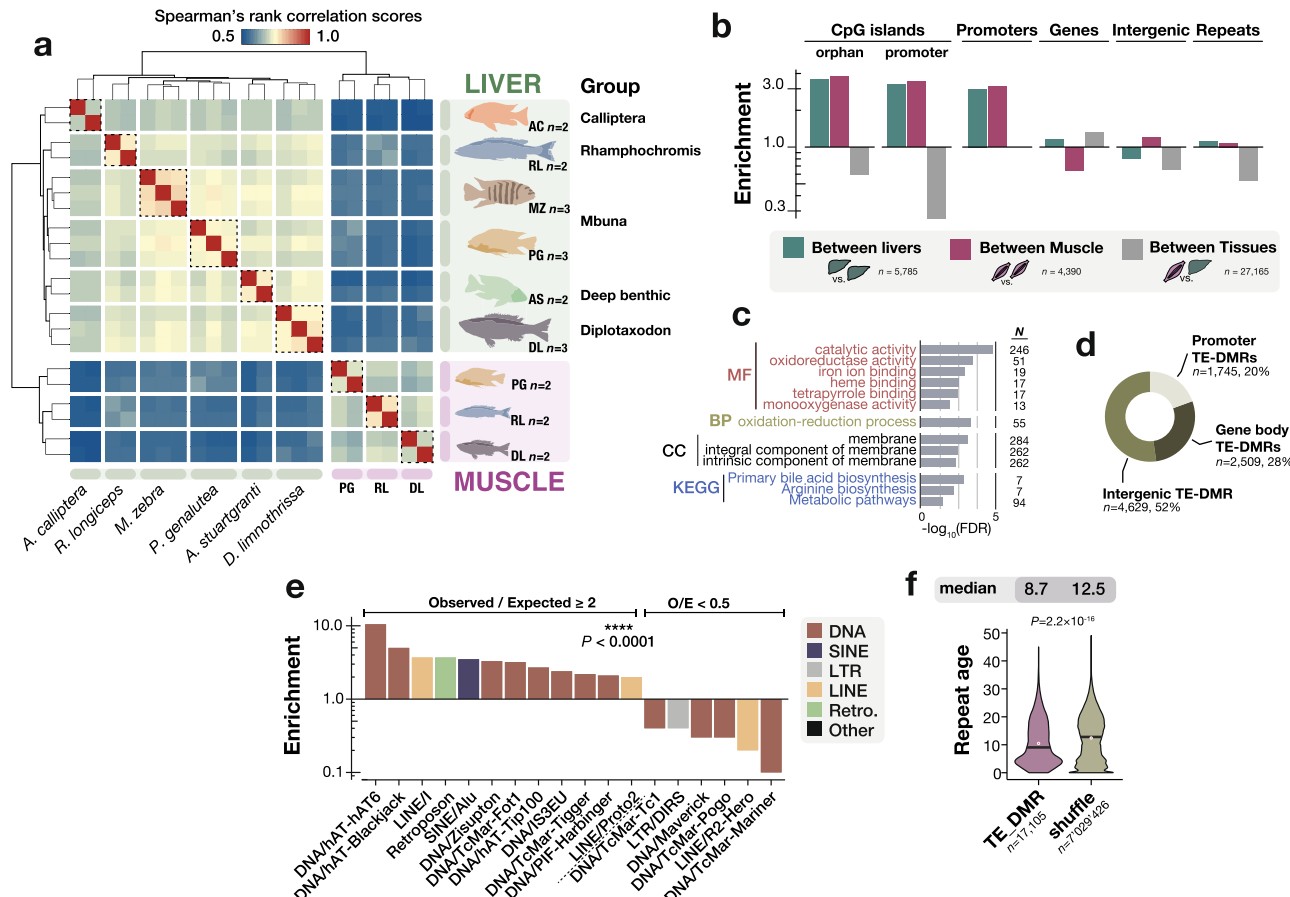

**Fig. 2 Species-specific methylome divergence in Lake Malawi cichlids is enriched in promoters, CpG-islands, and young transposons. a** Unbiased hierarchical clustering and heatmap of Spearman's rank correlation scores for genome-wide methylome variation in Lake Malawi cichlids at conserved CG dinucleotides. Dotted boxes group samples by species within each tissue. **b** Observed/Expected ratios (O/E ratio, enrichment) for some genomic localisations of differentially methylated regions (DMRs) predicted between livers (green) and between muscles (purple) of three Lake Malawi cichlid species, and between tissues (within-species, grey); $\chi^2$ tests for between categories ($p < 0.0001$), for O/E between liver and muscle DMRs ($p = 0.99$) and between Liver+Muscle vs Tissues ($p = 0.04$). Expected values were determined by randomly shuffling DMRs of each DMR type across the genome (1000 iterations). Categories are not mutually exclusive. **c** Gene ontology (GO) enrichment for DMRs found between liver methylomes localised in promoters. GO terms: Kyoto Encyclopaedia of Genes and Genomes (KEGG), molecular functions (MF), cellular component (CC), and biological processes (BP). Only GO terms with FDR < 0.05 shown. N indicates the number of genes associated with each GO term. Only GO terms with $p < 0.05$ (Benjamini −Hochberg false discovery rate [FDR]-corrected $p$-values) are shown. **d** Genomic localisation of liver DMRs containing repeats/transposons (TE-DMRs). **e**. O/E ratios for species TE-DMRs for each TE family. Only O/E ≥ 2 and ≤0.5 shown. $\chi^2$ tests, $p < 0.0001$. **f** Violin plots showing TE sequence divergence (namely, CpG-adjusted Kimura substitution level as given by RepeatMasker) in *M. zebra* genome for species TE-DMRs, TEs outside species DMRs ('outside') and randomly shuffled TE-DMRs (500 iterations, 'shuffle'). Mean values indicated by red dots, median values by black lines and shown above each graph. Total DMR counts indicated below each graph. Two-sided $p$-values for Kruskal–Wallis test are shown above the graph. DMR, differentially methylated region; TE, repeat/transposon regions; CGI, predicted CpG islands.

(Supplementary Fig. 9d), consistent with species-specific functional liver transcriptome activity.

Next, we checked for the association between liver DMRs and transcriptional changes. Of the 6,797 among-species DMRs that could be assigned to a specific gene (i.e., DMRs within promoters, gene bodies or located 0.5−4 kbp away from a gene; see "Methods"), 871 were associated with differentially expressed genes, which is greater than expected by chance (Fig. 3b; $p < 4.7 \times 10^{-5}$), suggesting that DMRs are significantly associated with liver gene expression. Of these 871 putative functional DMRs (pfDMRs), the majority (42.8%) are localised over gene bodies, hinting at possible intronic *cis*-regulatory elements or alternative splicing[49]. The remaining pfDMRs are in intergenic (30.2%) or promoters (27%) (Fig. 3c). The majority of pfDMRs contain younger TE sequences, in particular in intronic regions, while only few contain CGIs (Supplementary Fig. 10a−c). In promoters and intergenic regions, ≥63% of pfDMR sequences

contain TEs (Fig. 3c). As methylation levels at *cis*-regulatory regions may be associated with altered transcription factor (TF) activity[22,24,25], we performed TF binding motif enrichment analysis using between-species liver DMRs and found significant enrichment for specific TF recognition binding motifs. Several TF genes known to recognise some of the enriched binding motifs are differentially expressed among the livers of the three cichlid species and have liver-associated functions (Supplementary Fig. 10d, e). For example, the gene of the transcription factor hepatocyte nuclear factor 4 alpha (hnf4a), with important functions in lipid homeostasis regulation and in liver-specific gene expression[50], is >2.5x-fold downregulated ($q \leq 9 \times 10^{-5}$) in the rock-dwelling algae-eater *P. genalutea* compared to the pelagic piscivores *D. limnothrissa* and *R. longiceps*, possibly in line with adaptation to different diets (Supplementary Fig. 10e).

Furthermore, genomic regions containing pfDMRs are also significantly associated in the livers with altered transcription of

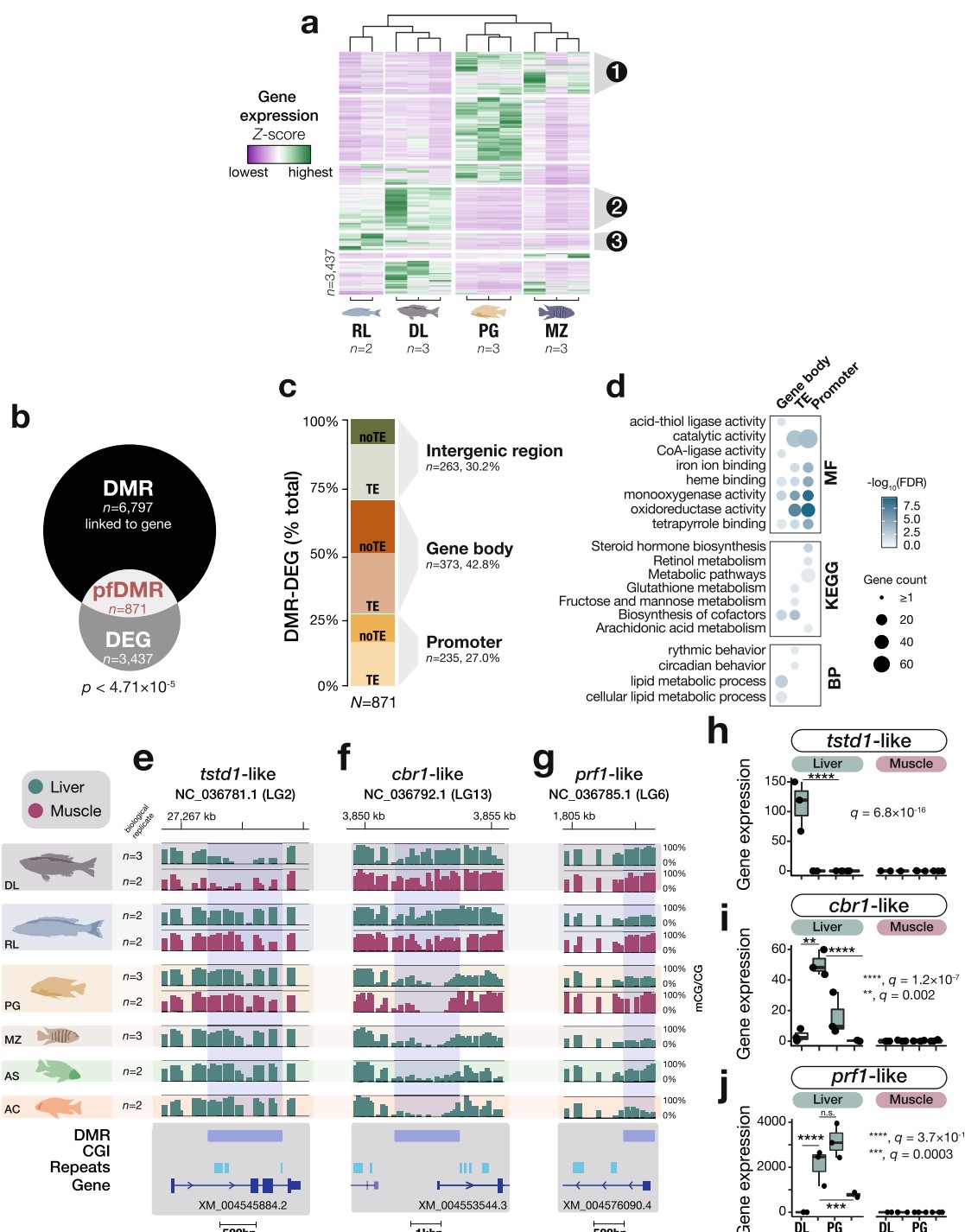

many other genes involved in hepatic and metabolic oxidation-reduction processes (Fig. 3d and Supplementary Fig. 10f). These include genes encoding haem-containing cytochrome P450 enzymes (such as cyp3a4, cy7b1; Supplementary Fig. 10f), which are important metabolic factors in steroid and fatty acid metabolism, as well as genes encoding other hepatic enzymes involved in energy balance processes. This enrichment is associated with significant methylome divergence among species, in particular in promoter regions and gene bodies (Fig. 3d). For example, the gene sulfurtransferase *tstd1*-like, an enzyme involved in energy balance and the mitochondrial metabolism, is expressed exclusively in the liver of the deep-water pelagic species *D. limnothrissa,* where it shows ~80% reduced methylation levels in

a gene-body DMR compared to all the other species (Fig. 3e, h). Another example is the promoter of the enzyme carbonyl reductase [NADPH] 1 (*cbr1*) which shows significant hypomethylation (2.2kbp-long DMR) in the algae-eaters MZ and PG, associated with up to ~60-fold increased gene expression in their livers compared to the predatory *Rhamphochromis* and *Diplotaxodon* (Fig. 3f, i). Interestingly, *cbr1* is involved in the metabolism of various fatty acids in the liver and has been associated with fatty acid-mediated cellular signalling in response to environmental perturbation[51]. As a final example, we highlight the cytotoxic effector perforin 1-like (*prf1*-like), an important player in liver-mediated energy balance and immune functions[52]. Its promoter is hypermethylated (>88% mCG/CG) exclusively in the

**Fig. 3 Methylome divergence is associated with differential transcriptional activity in Lake Malawi cichlids. a** Heatmap and unsupervised hierarchical clustering of gene expression values (Z-score) of all differentially expressed genes (DEGs) found among livers of four Lake Malawi cichlid species (Wald tests corrected for multiple testing using false discovery rate FDR < 1%). GO enrichment analysis for three DEG clusters are shown in Supplementary Fig. 9c. **b** Significant overlap between DEG and differentially expressed regions (DMRs; $p < 0.05$) linked to a gene (exact hypergeometric test, $p = 4.71 \times 10^{-5}$), highlighting putative functional DMRs (pfDMRs). **c** Bar plot showing the percentage of pfDMRs localised in either promoters, intergenic regions (0.5−4kbp away from genes), or in gene bodies, with the proportion of TE content for each group. **d** Heatmap representing significant GO terms for DEGs associated with pfDMRs for each genomic feature. GO categories: BP, Biological Process; MF, Molecular Function. Only GO terms with Benjamini−Hochberg FDR-corrected $p$-values < 0.05 are shown. Examples of pfDMRs significantly associated with species-specific liver transcriptional changes for the genes thiosulfate:glutathione sulfurtransferase tstd1-like (LOC101468457; $q = 6.82 \times 10^{-16}$) (**e**), carbonyl reductase [NADPH]-1 cbr1-like (LOC101465189; MZ vs DL, $q = 0.002$; MZ vs RL, $q = 1.18 \times 10^{-7}$) (**f**) and perforin-1 prf1-like (LOC101465185; MZ vs DL, $q = 3.68 \times 10^{-19}$; MZ vs RL, $q = 0.00034$) (**g**). Liver and muscle methylome profiles in green and purple, respectively (averaged mCG/CG levels [%] in 50 bp bins; $n = 3$ biological replicates for liver DL, PG, and MZ; $n = 2$ biological replicates for liver RL, AS, and AC, and muscle DL, RL, and PG). **h−j** Boxplots showing gene expression values (transcript per million) for the genes in (**e−g**). in livers (green) and muscle (pink). $n = 3$ biological replicates for liver DL, MZ, PG; $n = 2$ biological replicates for liver RL and muscle DL, MZ, PG, and RL. Two-sided $q$ values for Wald tests corrected for multiple testing (Benjamini−Hochberg FDR) are shown in graphs. Box plots indicate median (middle line), 25th, 75th percentile (box), and 5th and 95th percentile (whiskers) as well as outliers (single points). CGI, CpG islands; Repeats, transposons and repetitive regions.

liver of the deep-water species DL, while having low methylation levels (~25%) in the four other species (Fig. 3g). This gene is not expressed in DL livers but is highly expressed in the livers of the other species that all show low methylation levels at their promoters (Fig. 3j). Taken together, these results suggest that species-specific methylome divergence is associated with transcriptional remodelling of ecologically-relevant genes, which might facilitate phenotypic diversification associated with adaption to different diets.

**Multi-tissue methylome divergence is enriched in genes related to early development.** We further hypothesised that between-species DMRs that are found in both the liver and muscle methylomes could relate to functions associated with early development/embryogenesis. Given that liver is endoderm-derived and muscle mesoderm-derived, such shared multi-tissue DMRs could be involved in processes that find their origins prior to or early in gastrulation. Such DMRs could also have been established early on during embryogenesis and may have core cellular functions. Therefore, we focussed on the three species for which methylome data were available for both tissues (Fig. 1c) to explore the overlap between muscle and liver DMRs (Fig. 4a). Based on pairwise species comparisons (Supplementary Fig. 11a, b), we identified methylome patterns unique to one of the three species. We found that 40−48% of these were found in both tissues ('multi-tissue' DMRs), while 39−43% were liver-specific and only 13−18% were muscle-specific (Fig. 4b).

The relatively high proportion of multi-tissue DMRs suggests there may be extensive among-species divergence in core cellular or metabolic pathways. To investigate this further, we performed GO enrichment analysis. As expected, liver-specific DMRs are particularly enriched for hepatic metabolic functions, while muscle-specific DMRs are significantly associated with muscle-related functions, such as glycogen catabolic pathways (Fig. 4c). Multi-tissue DMRs, however, are significantly enriched for genes involved in development and embryonic processes, in particular related to cell differentiation and brain development (Fig. 4c−f), and show different properties from tissue-specific DMRs. Indeed, in all the three species, multi-tissue DMRs are three times longer on average (median length of multi-tissue DMRs: 726 bp; Dunn's test, $p < 0.0001$; Supplementary Fig. 11c), are significantly enriched for TE sequences (Dunn's test, $p \leq 0.03$; Supplementary Fig. 11d) and are more often localised in promoter regions (Supplementary Fig. 11e) compared to liver and muscle DMRs. Furthermore, multi-tissue species-specific methylome patterns

show significant enrichment for specific TF binding motif sequences. These binding motifs are bound by TFs with functions related to embryogenesis and development, such as the transcription factors Forkhead box protein K1 (foxk1) and Forkhead box protein A2 (foxa2), with important roles during liver development[53] (Supplementary Fig. 11f), possibly facilitating core phenotypic divergence early on during development.

Several examples of multi-tissue DMRs are worth highlighting as generating hypotheses for potential future functional studies (Fig. 4d−f). The visual system homeobox 2 (vsx2) gene in the offshore deep-water species *Diplotaxodon limnothrissa* is almost devoid of methylation in both liver and muscle, in contrast to the other species (1.9 kbp-long DMR; Fig. 4d and Supplementary Fig. 11g). vsx2 has been reported to play an essential role in the development of the eye and retina in zebrafish with embryonic and postnatal active transcription localised in bipolar cells and retinal progenitor cells[54]. *D. limnothrissa* populates the deepest parts of the lake of all cichlid species (down to approximately 250 m, close to the limits of oxygenation) and features morphological adaptations to dimly-lit environments, such as larger eye size[55]. vsx2 may therefore participate in the visual adaptation of *Diplotaxodon* to the dimmer parts of the lake via DNA methylation-mediated gene regulation during development. Another example of a multi-tissue DMR specific to *D. limnothrissa* is located in the promoter of the gene coding for the growth-associated protein 43 (gap43) involved in neural development and plasticity, and also neuronal axon regeneration[56]. The promoter of gap43 is largely devoid of methylation (overall <5% average mCG/CG levels over this 5.2 kbp-long DMR) in both muscle and liver tissues of *D. limnothrissa*, while being highly methylated (>86% mCG/CG) in the other species (Fig. 4e). In *A. calliptera*, the transcription of gap43 is restricted to the brain and embryo (Supplementary Fig. 11h), consistent with a role in neural development and in the adult brain. Finally, another multi-tissue DMR potentially involved in neural embryonic functions is located in the promoter region of the gene tenm2, coding for teneurin transmembrane protein (Fig. 4f). tenm2 is a gene expressed early on during zebrafish embryogenesis as well as in cichlid brain and embryo (Supplementary Fig. 11i) and is involved in neurodevelopment and neuron migration-related cell signalling[57]. This 2.7 kbp-long DMR is completely unmethylated in the algae-eating rock-dweller *Petrotilapia genalutea* (almost 80% reduction in methylation levels overall compared to the other species) and may mediate species-specific adaptive phenotypic plasticity related to synapse formation and neuronal networks.

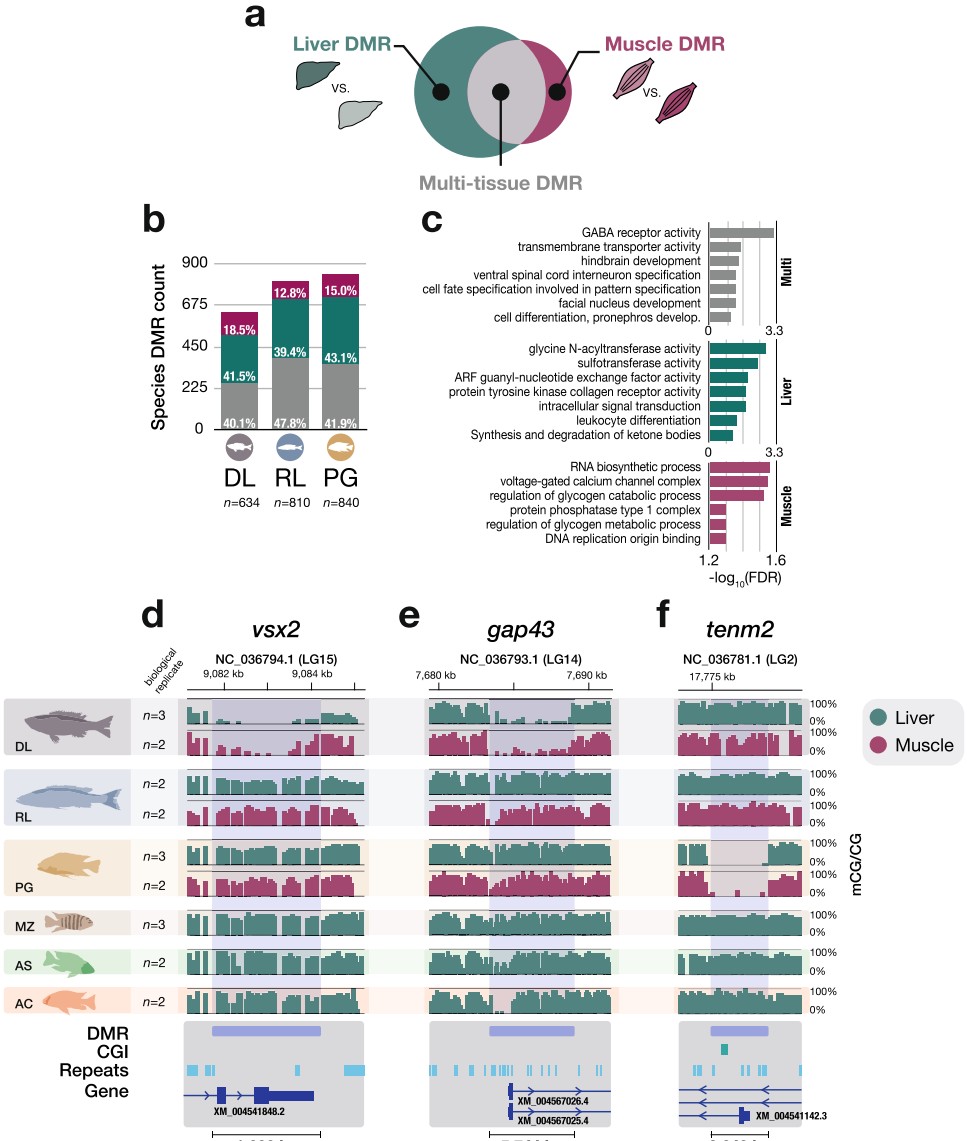

**Fig. 4 Multi-tissue methylome divergence in Lake Malawi cichlids is associated with early development/embryogenesis. a** Distinct species-specific methylome patterns in Lake Malawi cichlids can be found in liver or muscle tissues, or in both tissues ('multi-tissue'). **b** Histograms showing the total counts of 'species' DMRs that are either liver-, muscle-specific or present in both (multi). Only 'species' DMRs showing distinct DNA methylation patterns in one species are shown. **c** GO enrichment plots for each DMR class. Only GO terms with Benjamini—Hochberg FDR-corrected *p*-values < 0.05 are shown. **d**—**f** Examples of 'species' multi-tissue DMRs in genes related to embryonic and developmental processes. Namely, in the genes coding for visual system homeobox 2 *vsx2* (LOC101486458), growth-associated protein 43 *gap43* (LOC101472990) and teneurin transmembrane protein 2 *tenm2* (LOC101470261). Liver and muscle methylome profiles shown in green and purple, respectively (averaged mCG/CG levels [%] in 50 bp bins for ≥2 samples per tissue per species; scale indicated below each graph).

## Discussion

The molecular mechanisms underlying adaptive phenotypic diversification are subject of intense interest[34,36,38,58,59] and the extent of the role of epigenetic processes is hotly debated[2,4,60]. However, in-depth molecular epigenetic studies remain rare in evolutionary genomics and its key model systems[2,4,29,60]. Here, we focussed on the genetically closely related haplochromine cichlids of Lake Malawi, representing a unique system to investigate the epigenetic basis for phenotypic diversification[36,39,61]. Specifically, we describe genome-wide methylome variation at a single CG dinucleotide resolution as well as transcriptomes of two adult tissues of different embryonic origins in eco-morphologically divergent species (Fig. 1b). This work investigates epigenetic marks in the context of rapid diversification in

natural populations of cichlid fishes and provides evidence of substantial methylome divergence associated with ecologically-relevant genes and correlated with changes in the transcriptional network and in TF activity. Given the resemblances we found between cichlid methylomes and those of warm-blooded vertebrates (Fig. 1d, e), suggesting evolutionarily conserved functions, our findings are likely to be relevant to other vertebrate evolutionary model systems.

Recent large-scale epigenetic studies in natural populations of *Arabidopsis* have highlighted a functional link between local environments and methylation divergence, with possible adaptive phenotypic functions[11,13]. Yet, epigenetic variation in natural populations of vertebrates and its possible functions in the context of adaptive phenotypic diversification have scarcely been

studied. Our finding of considerable among-species methylome divergence at conserved underlying DNA sequences, despite overall low among-species genome differentiation, is suggestive of a functional link between DNA methylation and local environments, which may facilitate phenotypic plasticity and diversification. The methylome divergence we found may be driven directly by environmental differences but is also likely to have a genetic component. Our study lays the groundwork for deciphering any genetically encoded component underlying the epigenetic differences. Genetic differences in TF binding domains or in TF sequence recognition motifs, as well as in the proteins involved in the maintenance and deposition of new methyl groups, could for example lead to epigenetic divergence[11,24]. While this study provides evidence for species-specific methylome divergence associated with transcriptional changes of ecologically-relevant genes, further experimental work is required to examine the extent to which such species-specific patterns have an adaptive function in a natural context, as well as to determine the degree of plasticity and inheritance of such epigenetic patterns. Recent studies in three-spined stickleback fish have provided initial evidence for stable transmission of methylome patterns across generations associated with adaptation to salinity, some of which are inherited in a genetic-independent manner[62,63]. Moreover, epigenetic inheritance and reprogramming greatly vary among teleost fishes. Indeed, recent studies have highlighted important differences in the mechanisms of DNA methylation reprogramming during embryogenesis in teleost fishes. While the genome of the embryo in zebrafish retains the sperm methylome configuration with no global DNA methylation resetting, possibly allowing for the transgenerational inheritance of specific epigenetic states, extensive and global DNA methylation reprogramming instead occurs upon fertilisation in medaka embryos (similar to mammals)[30,64–66]. Such DNA methylome reprogramming processes are currently unknown in cichlids, which warrants further research.

We found that regions of methylome divergence between species (DMRs) were enriched in promoters and orphan CGIs (Fig. 2b). Methylation variation in promoter regions is known to have important cis-regulatory functions in vertebrates, in particular during development[20,21,24,29,31]. Such cis-regulatory activity is also apparent in Lake Malawi cichlids, with methylation at promoters negatively correlated with transcriptional activity (Fig. 1e and Supplementary Fig. 7a–d). This is likely mediated by the tight interaction of DNA methylation with 5mC-sensitive DNA-binding proteins, such as many transcription factors[22] (see below). On the other hand, the functional roles of orphan CGIs are less well understood[42]. However, orphan CGIs have by far the highest enrichment for species methylome divergence (>3-fold over chance; Fig. 2b)—most of which are located in unannotated genomic regions. Orphan CGIs, as well as intergenic TEs (Fig. 2d), might include ectopic promoters, enhancers and other distal regulatory elements[41,42] that may participate in phenotypic diversification by reshaping transcriptional network. Such putative cis-regulatory regions could be validated against a full functional annotation of the genome of Lake Malawi cichlid, which is currently lacking.

We identified that in some species methylome divergence was significantly associated with differential liver transcriptome activity, especially pertaining to hepatic functions involved in steroid hormone and fatty acid metabolism (Fig. 3b, d–j). Consistent with a functional role of DNA methylation in cis-regulatory regions[21,44], we revealed significant methylation divergence in the promoters of differentially transcribed genes involved in liver-mediated energy expenditure processes and metabolism, such as gene prf1-like (>60-fold increase in expression; Fig. 3g, j), associated with obesity in mouse[44]. Such a

functional link may promote phenotypic diversification via adaptation to different diets. Our understanding of this would benefit from the knowledge of the extent to which environmental or diet perturbation might result in adaptation-associated functional methylome changes. Further work would also be required to assess the extent to which such changes may be stably inherited. Additionally, the characterisation of the methylomes of Lake Malawi cichlid species from different ecomorphological groups but sharing the same habitat/diet, would inform on the specificity and possible functions of methylome divergence at metabolic genes.

We observed that methylome divergence associated with altered transcription in livers is enriched for binding motifs recognised by specific TFs. Some of those TFs are also differentially expressed in the livers and have important roles in lipid and energy homeostasis (Supplementary Fig. 10d, e). This suggests that altered activity of some TFs in livers can be associated with species-specific methylome patterns. Methylome variation in cis-regulatory regions is known to affect the binding affinity of methyl-sensitive DNA-binding regulatory factors (such as TFs)[25,44,67,68]. Furthermore, methylation-associated changes in chromatin accessibility may also impede the binding affinity of such factors and could be associated with altered TF activity and changes in transcription[20,67]. Alternatively, altered TF activity, arising from species-specific mutations within TF binding sequence motifs or in TF binding domains, has also been reported to generate methylome divergence in cis and trans[24], and could also underlie species-specific epigenetic divergence. Our results suggest a tight link between TF activity and methylome divergence, that could participate in reshaping the transcriptional network of the livers in Lake Malawi cichlids.

TE and repetitive sequences present on average higher methylation levels than the genome-wide average (Fig. 1d), although some specific TE classes show more variable and lower levels (Supplementary Fig. 6d, e). DNA methylation-mediated transcriptional repression of mostly deleterious TE elements is crucial to the integrity of most eukaryote genomes, from plants to fish and mammals, and can be mediated in both animals and plants by small non-coding RNAs, such as piwi-interacting RNAs (piRNAs) in zebrafish and mammals[18,19,69]. Notably, the majority (~60%) of species differences in methylation patterns associated with transcriptional changes in liver was significantly localised in evolutionary young transposon/repeat regions, notably in intergenic retroposons in the vicinity of genes and in intronic DNA transposons (Dunn's test $p < 10^{-10}$; Fig. 3c and Supplementary Fig. 10b). Even though most of TE activity is under tight cellular control to ensure genome stability, transposition events have also been associated with genome evolution and phenotypic diversification. Indeed, TE insertion may represent a source of functional genomic variation and novel cis-regulatory elements, underlying altered transcriptional network[45,47,48,70]. In haplochromine cichlids, variation in anal fin egg-spots patterns associated with courtship behaviour, has been linked to a novel cis-regulatory element, derived from TE sequences[46]. Additionally, Brawand and colleagues have revealed that most TE insertions near genes in East African cichlids were associated with altered gene expression patterns[38]. Moreover, genes in piRNA-related pathways have been reported to be under positive selection in Lake Malawi cichlid flock, in line with a fast evolving TE sequence landscape observed in cichlids[36], and these genes may also be associated with TE-related methylome variation, similar to Arabidopsis[11,71].

Not only can novel TE insertions participate in genome evolution, DNA methylation at TE-derived cis-regulatory elements has been shown to affect transcriptional activity of nearby genes[12,45]. In rodents, the insertion of one IAP (intra-cisternal A

particle) retrotransposon in the upstream *cis*-regulatory region of the agouti gene is associated with considerable phenotypic variation of coat colours and metabolic changes. Differential methylation levels at this TE-derived ectopic promoter directly impacts the activity of the agouti gene[5,28], and such epigenetic patterns of methylation are transmitted to the offspring along with the altered phenotypes in a non-genetic manner[2]. Similarly, in toadflax, the flower symmetry is associated with the variable and heritable methylation patterns in the TE-derived promoter of the *Lcyc* gene, resulting in symmetrical or asymmetrical flowers[6]. Also, in a population-scale study of more than a thousand natural *Arabidopsis* accessions, epigenetic variation was found to be associated with phenotypes, mostly arising from methylation-mediated TE silencing that was significantly associated with altered transcription of adaptive genes such as those determining flowering time[11,71]. Our work adds to this by providing further evidence that interactions between TE sequences and between-species methylome divergence might have led to altered transcriptional networks. This lays the groundwork for further investigation of this issue in cichlid fishes.

Finally, we revealed that between-species methylome differences in liver tissues were greater than differences between muscle tissues (Fig. 4b), possibly highlighting a higher dependence of hepatic functions on natural epigenetic divergence. This indicates that a significant portion of the between-species methylome divergence in the liver may be associated with phenotypic divergence, in particular by affecting genes involved in tissue-specific functions, such as hepatic metabolic processes (Fig. 3c, e–j). However, almost half of the methylome divergence we observed that was driven by a single species was consistently found in both liver and muscle (Fig. 4b). This multi-tissue methylome divergence is consistent with epigenetic influences on core cellular functions and may also be relevant to early-life biological processes such as development, cellular differentiation, and embryogenesis (Fig. 4c, d–f). For example, we identified a large hypomethylated region in the visual homeobox gene *vsx2* in both liver and muscle tissues in the deep-water *Diplotaxodon* (Fig. 4d). This gene is involved in eye differentiation and may participate in long-lasting visual phenotypic divergences required to populate dimly parts of the lake, similar to the DNA methylation-mediated adaptive eye degeneration in cavefish[29]. Notably, recent studies have highlighted signatures of positive selection and functional substitutions in genes related to visual traits in *D. limnothrissa*[36,55]. Furthermore, in regions showing multi-tissue species-specific methylome divergence, we identified significant enrichment for binding motifs of specific TFs whose functions are related to embryogenesis and liver development (such as foxa2 and foxk1). This suggests that altered TF activity during development could be associated with species-specific methylome patterns (Supplementary Fig. 11f). If multi-tissue methylome divergence has been established very early during differentiation, and has important regulatory functions pertaining to early developmental stages[26] and possibly core cellular functions, then it may promote long-lasting phenotypic divergence unique to each species' adaptions. Our observations suggest that further characterisation of the methylomes and transcriptomes of different cells of the developing embryo may be valuable to investigate when between-species methylome divergence is established, as well as any functional roles in early-life phenotypic diversification.

To conclude, recent large-scale genomic studies have highlighted that several mechanisms may participate in the phenotypic diversification of Lake Malawi haplochromine cichlids, such as hybridisation and incomplete lineage sorting[34,36,61,72]. Our study adds to these observations by providing initial evidence of substantial methylome divergence associated with altered

transcriptome activity of ecologically-relevant genes among closely related Lake Malawi cichlid fish species. This raises the possibility that variation in methylation patterns could facilitate phenotypic divergence in these rapidly evolving species through different mechanisms (such as altered TF binding affinity, gene expression, and TE activity, all possibly associated with methylome divergence at *cis*-regulatory regions). Further work is required to elucidate the extent to which this might result from plastic responses to the environment and the degree of inheritance of such patterns, as well the adaptive role and any genetic basis associated with epigenetic divergence. This study represents an epigenomic study investigating natural methylome variation in the context of phenotypic diversification in genetically similar but ecomorphologically divergent cichlid species part of a massive vertebrate radiation and provides an important resource for further experimental work.

## Methods

**Sampling overview.** All cichlid specimens were bought dead from local fishermen by G.F. Turner, M. Malinsky, H. Svardal, A.M. Tyers, M. Mulumpwa, and M. Du in 2016 in Malawi in collaboration with the Fisheries Research Unit of the Government of Malawi), or in 2015 in Tanzania in collaboration with the Tanzania Fisheries Research Institute (various collaborative projects). Sampling collection and shipping were approved by permits issued to G.F. Turner, M.J. Genner R. Durbin, E.A. Miska by the Fisheries Research Unit of the Government of Malawi and the Tanzania Fisheries Research Institute, and were approved and in accordance with the ethical regulations of the Wellcome Sanger Institute, the University of Cambridge and the University of Bangor (UK). Upon collection, tissues were immediately placed in RNA*later* (Sigma) and were then stored at −80 °C upon return. Information about the collection type, species IDs, and the GPS coordinates for each sample in Supplementary Data 1.

**SNP-corrected genomes.** Because real C > T (or G > A on the reverse strand) mutations are indistinguishable from C > T SNPs generated by the bisulfite treatment, they can add some bias to comparative methylome analyses. To account for this, we used SNP data from Malinsky et al. (2018) (ref. [36]) and, using the *Maylandia zebra* UMD2a reference genome (NCBI_Assembly: GCF_000238955.4) as the template, we substituted C > T (or G > A) SNPs for each of the six species analysed before re-mapping the bisulfite reads onto these 'updated' reference genomes.

To translate SNP coordinates from Malinsky et al. (2018) to the UMD2a assembly, we used the UCSC liftOver tool (version 418), based on a whole genome alignment between the original Brawand et al., 2014 (ref. [38]) (https://www.ncbi.nlm.nih.gov/assembly/GCF_000238955.1/) and the UMD2a *M. zebra* genome assemblies. The pairwise whole genome alignment was generated using lastz v1.02[73], with the following parameters: "B = 2 C = 0 E = 150 H = 0 K = 4500 L = 3000 M = 254 O = 600 Q = human_chimp.v2.q T = 2 Y = 15000". This was followed by using USCS genome utilities (https://genome.ucsc.edu/util.html) axtChain (kent source version 418) tool with -minScore=5000. Additional tools with default parameters were then used following the UCSC whole-genome alignment paradigm (http://genomewiki.ucsc.edu/index.php/Whole_genome_alignment_howto) in order to obtain a contiguous pairwise alignment and the 'chain' file input for liftOver (kent source version 418).

The 'lifted over' C > T (or G > A) SNPs were then substituted into the UMD2a genome using the evo getWGSeq command with the–whole-genome and–methylome options. The code used is available as a part of the Evo package (https://github.com/millanek/evo; v.0.1 r24, commit99d5b22).

**Extraction of high-molecular-weight genomic DNA (HMW-gDNA).** The main method to generate WGBS data is summarised in Supplementary Fig. 1. In detail, high-molecular-weight genomic DNA (HMW-gDNA) was extracted from homogenised liver and muscle tissues (<25 mg) using QIAamp DNA Mini Kit (Qiagen 51304) according to the manufacturer's instructions. Before sonication, unmethylated lambda DNA (Promega, D1521) was spiked in (0.5% w/w) to assess bisulfite conversion efficiency. HMW-gDNA was then fragmented to the target size of ~400 bp (Covaris, S2, and E220). Fragments were then purified with PureLink PCR Purification kit (ThermoFisher). Before any downstream experiments, quality and quantity of gDNA fragments were both assessed using NanoDrop, Qubit, and Tapestation (Agilent).

**Sequencing library preparation—whole-genome bisulfite sequencing.** For each sample, 200 ng of sonicated fragments were used to make NGS (next-generation sequencing) libraries using NEBNext Ultra II DNA Library Prep (New England BioLabs, E7645S) in combination with methylated adaptors (NEB, E7535S),

following the manufacturer's instructions. Adaptor-ligated fragments were then purified with 1.0x Agencourt AMPure Beads (Beckman Coulter, Inc). Libraries were then treated with sodium bisulfite according to the manufacturer's instructions (Imprint DNA Modification Kit; Sigma, MOD50) and amplified by PCR (10 cycles) using KAPA HiFi HS Uracil+ RM (KAPA Biosystems) and NEBNext Multiplex Oligos for Illumina (NEB E7335S). Bisulfite-converted libraries were finally size-selected and purified using 0.7x Agencourt AMPure Beads. The size and purity of libraries were determined using Tapestation and quantified using Qubit (Agilent). Whole-genome bisulfite sequencing (WGBS) libraries were sequenced on HiSeq 4000 (High Output mode, v.4 SBS chemistry) to generate paired-end 150 bp-long reads. *A. stuartgranti* samples were sequenced on HiSeq 2500 to generate paired-end 125 bp-long reads.

**Mapping of WGBS reads.** TrimGalore (options: --paired --fastqc --illumina; v0.6.2; github.com/FelixKrueger/TrimGalore) was used to determine the quality of sequenced read pairs and to remove Illumina adaptor sequences and low-quality reads/bases (Phred quality score < 20). All adaptor-trimmed paired reads from each species were then aligned to the respective species-specific SNP-corrected *M.zebra* genomes (see above and Supplementary Data 1) and to the lambda genome (to determine bisulfite non-conversion rate) using Bismark[74] (v0.20.0). The alignment parameters were as follows: 0 mismatch allowed with a maximum insert size for valid paired-end alignments of 500 bp (options: -p5 -N 0 –X 500). Clonal mapped reads (i.e., PCR duplicates) were removed using Bismark's deduplicate_bismark (see Supplementary Data 1). Mapped reads for the same samples generated on multiple HiSeq runs were also merged.

**Differentially methylated regions (DMR) and comparative analysis.** Methylation at CpG sites was called using Bismark's bismark_methylation_extractor (options: -p --multicore 9 --comprehensive --no_overlap --merge_non_CpG). DMRs (>25% methylation difference, ≥50 bp, ≥4 CG and $p < 0.05$) were predicted using DSS[75] (v2.32.0). samtools (v1.9) and bedtools (v2.27.1) were used to generate averaged methylation levels across non-overlapping windows of various sizes genome-wide. ggplot2 (v3.3.0) and pheatmap (v1.0.12) were used to visualise methylome data and to produce unbiased hierarchal clustering (Euclidean's distances and complete-linkage clustering). Spearman's correlation matrices, Euclidean distances, and principal component analyses (scaled and centred) were produced using R (v3.6.0) functions cor, dist, and prcom, respectively. The minimum read overage requirement at any CpG sites for all analyses—except for DSS-predicted DMRs, for which all read coverage was used—was as follows: >4 and ≤100 non-PCR-duplicate mapped paired-end reads. mCG % levels over 50 bp-long non-overlapping windows for all annotations were averaged for each tissue of each sample. The genome browser IGV (v2.5.2) was used to visualise DNA methylation levels genome-wide (% mCG/CG in 50 bp windows; bigwig format).

**Additional statistics.** Kruskal−Wallis H and Dunn's multiple comparisons tests (using Benjamini−Hochberg correction, unless otherwise specified) were performed using FSA (v0.8.25). Box plots indicate median (middle line), 25th, 75th percentile (box), and 5th and 95th percentile (whiskers) as well as outliers (single points). Violin plots were generated using ggplot2 and represent rotated and mirrored kernel density plots.

**Genomic annotations.** The reference genome of *M. zebra* (UMD2a; NCBI genome build: GCF_000238955.4 and NCBI annotation release 104) was used to generate all annotations. Custom annotation files were generated and were defined as follows: promoter regions, TSS ± 500 bp unless otherwise indicated; gene bodies included both exons and introns and other intronic regions, and excluded the first 500 bp regions downstream of TSS to avoid any overlap with promoter regions; transposable elements and repetitive elements (TE) were modelled and annotated, as well as their sequence divergence analysed, using RepeatModeler (v1.0.11) and RepeatMasker (v4.0.9.p2), respectively. Intergenic regions were defined as genomic regions more than 0.5 kbp away from any gene. CpG-rich regions, or CpG islands (CGI), were predicted and annotated using makeCGI (v1.3.4)[76]. The following genomes were used to compare genomic CG contents across different organisms (Supplementary Fig. 5a): honey bee (*A. melifera*, Amel_4.5), nematode (*C. elegans*, WBcel235), *Arabidopsis* (*A. thaliana*, TAIR10), zebrafish (*D. rerio*, GRCz10), Mbuna cichlid *Maylandia zebra* (*M. zebra*, UMD1), West Indian Ocean coelacanth (*L. chalumnae*, LatCha.1), red junglefowl (*G. gallus*, Gall_5), grey whale (*E. robustus*, v1), human (*H. sapiens*, GRCh38.p10), mouse (*M. musculus*, GRCm38.p5), tammar wallaby (*N. eugenii*, Meug1.1). pfDMRs and transposon/repeat elements were assigned to a gene when they were located within gene bodies (from 0.5 kbp downstream TSS), within promoter regions (TSS ± 500 bp) and in the vicinity of genes (0.5−4 kbp away from genes).

**Enrichment analysis.** Enrichment analysis was calculated by shuffling each type of DMRs (liver, muscle, tissue) across the *M.zebra* UMD2a genome (accounting for the number of DMRs and length; 1000 iterations). The expected values were determined by intersecting shuffled DMRs with each genomic category. Chi-square tests were then performed for each Observed/Expected (O/E) distribution. The same process was performed for TE enrichment analysis.

**Gene Ontology (GO) enrichment analysis.** All GO enrichment analyses were performed using g:Profiler (https://biit.cs.ut.ee/gprofiler/gost; version: e104_eg51_p15_3922dba [September 2020]). Only annotated genes for *Maylandia zebra* were used with a statistical cut-off of FDR < 0.05 (unless otherwise specified).

**Sequence divergence.** A pairwise sequence divergence matrix was generated using a published dataset[36]. Unrooted phylogenetic trees and heatmap were generated using the following R packages: phangorn (v.2.5.5), ape_5.4-1 and pheatmap (v.1.0.12).

**Total RNA extraction and RNA sequencing.** In brief, for each species, 2-3 biological replicates of liver and muscle tissues were used to sequence total RNA (see Supplementary Fig. 1 for a summary of the method and Supplementary Table 1 for sampling size). The same specimens were used for both RNAseq and WGBS.

RNAseq libraries for both liver and muscle tissues were prepared using ~5−10 mg of RNA*later*-preserved homogenised liver and muscle tissues. Total RNA was isolated using a phenol/chloroform method following the manufacturer's instructions (TRIzol, ThermoFisher). RNA samples were treated with DNase (TURBO DNase, ThermoFisher) to remove any DNA contamination. The quality and quantity of total RNA extracts were determined using NanoDrop spectrophotometer (ThermoFisher), Qubit (ThermoFisher), and BioAnalyser (Agilent). Following ribosomal RNA depletion (RiboZero, Illumina), stranded rRNA-depleted RNA libraries (Illumina) were prepped according to the manufacturer's instructions and sequenced (paired-end 75bp-long reads) on HiSeq2500 V4 (Illumina) by the sequencing facility of the Wellcome Sanger Institute. Published RNAseq dataset[36] for all *A. calliptera* sp. Itupi tissues were used (NCBI Short Read Archive BioProjects PRJEB1254 and PRJEB15289).

**RNAseq reads mapping and gene quantification.** TrimGalore (options: --paired --fastqc --illumina; v0.6.2; https://github.com/FelixKrueger/TrimGalore) was used to determine the quality of sequenced read pairs and to remove Illumina adaptor sequences and low-quality reads/bases (Phred quality score <20). Reads were then aligned to the *M. zebra* transcriptome (UMD2a; NCBI genome build: GCF_000238955.4 and NCBI annotation release 104) and the expression value for each transcript was quantified in transcripts per million (TPM) using kallisto[77] (options: quant --bias -b 100 -t 1; v0.46.0). For all downstream analyses, gene expression values for each tissue were averaged for each species.

To assess transcription variation across samples, a Spearman's rank correlation matrix using overall gene expression values was produced with the R function cor. Unsupervised clustering and heatmaps were produced with R packages ggplot2 (v3.3.0) and pheatmap (v1.0.12; see above). Heatmaps of gene expression show scaled TPM values (Z-score).

**Differential gene expression (DEG) analysis.** Differential gene expression analysis was performed using sleuth[78] (v0.30.0; Wald test, false discovery rate adjusted two-sided *p*-value, using Benjamini−Hochberg <0.01). Only DEGs with gene expression difference of ≥50 TPM between at least one species pairwise comparison were analysed further.

**Correlation between methylation variation and differential transcriptional activity.** To study the correlation between methylome and gene expression levels (Fig. 1e and Supplementary Fig. 7), genes were binned into 11 categories based on their expression levels (increasing gene expression levels, from category 1 to 10); cat "OFF" grouped silent/not expressed genes, i.e., TPM = 0 in all replicates for a particular species. RL liver ($n = 2$ biological replicates): 10 'ON' categories, $n = 2,129$ each; 1 'OFF' category, $n = 5,331$. MZ liver ($n = 3$ biological replicates): 10 'ON' categories, $n = 2,199$ each; 1 'OFF' category, $n = 4,704$. RL muscle ($n = 2$ biological replicates): 10 'ON' categories, $n = 2,101$ each; 1 'OFF' category, $n = 4,622$. Promoters (500 bp ± TSS) and gene bodies were also binned into 10 categories according to methylation levels (0−100% average methylation levels, by 10% DNA methylation increment); RL liver ($n = 2$ biological replicates), 11 categories, $n$ ranging from 34 to 11,202 per category. MZ liver ($n = 3$ biological replicates), 11 categories, $n$ ranging from 28 to 11,192 per category. RL muscle ($n = 2$ biological replicates), 11 categories, $n$ ranging from 60 to 9,946 per category. Categories were generated using the R script tidyverse (v1.3.0) and graphs were generated using deepTools v.3.2.1. TPM values and methylation levels were averaged for each tissue and each species.

**Reporting summary.** Further information on research design is available in the Nature Research Reporting Summary linked to this article.

## Data availability
The data that support this study are available from the corresponding authors upon reasonable request. All raw sequencing reads (WGBS, RNAseq, and SNP-corrected genomes), and processed data generated in the course of this study have been deposited in the Gene Expression Omnibus (GEO) database under the accession number GSE158514. Sample accessions are listed in Supplementary Data 1. In addition, variant call files (for SNP-corrected genomes and pairwise whole-genome sequence divergence),

as well as RNAseq for *A. calliptera* tissues were downloaded from NCBI Short Read Archive BioProjects PRJEB1254 and PRJEB15289. The source data are provided with this paper.

## Code availability

The code used to generate SNP-substituted genomes is available as a part of the Evo package (https://github.com/millanek/evo; v.0.1 r24, commit99d5b22).

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

## Acknowledgements

We would like to thank S.M. Grant's diving team for collecting some of the fish specimens, as well as the Fisheries Research Unit of the Government of Malawi, and the Tanzania Fisheries Research Institute, for their assistance and support. We would like to thank the staff at the Gurdon Institute and the sequencing facilities at CRUK Cambridge Institute, Gurdon, and Sanger Institutes for their expertise and support. We would like to thank the members of the Miska Lab as well as the Cambridge Cichlid community for fruitful discussions. We thank Navin B. Ramakrishna for critical comments on the manuscript, as well as David Jordan, Tomás di Domenico and Konrad L.M. Rudolph for their support with data analysis. We are grateful to Ole Seehausen and Marcel Häsler (University of Bern, Switzerland) for providing PN tissues. We thank Alan Hudson for help with sample collection. The map of Africa (Fig. 1a) was downloaded and modified from https://www.d-maps.com/carte.php?num_car=733&lang=en. This work was supported by the following grants to E.A.M.: Wellcome Trust Senior Investigator Award (104640/Z/14/Z and 219475/Z/19/Z) and CRUK award (C13474/A27826); to R.D.: Wellcome award (WT207492); to G.F.T. and M.J.G., the Leverhulme Trust—Royal Society Africa Awards (AA100023 and AA130107), and NERC award (NE/S001794/1). G.V. thanks Wolfson College, University of Cambridge, and the Genetics Society, London for financial support. The authors also acknowledge core funding to the Gurdon Institute from Wellcome (092096/Z/10/Z, 203144/Z/16/Z) and CRUK (C6946/A24843). For Open Access, the author has applied a CC BY public copyright licence to any Author Accepted Manuscript version arising from this submission.

## Author contributions

G.V. and E.A.M. devised and supervised the study; G.V. and E.A.M. interpreted the results with contributions from G.F.T., M.J.G, M.E.S., M.M. and R.D.; G.V. performed all experiments and analyses with contribution from M.M. (SNP-corrected genomes); M.D., H.S., M.M., R.D., M.J.G., G.F.T., and A.M.T. collected and provided tissues from wild-caught cichlid specimens; M.D. extracted total RNA samples; G.V. and E.A.M. wrote the paper with comments and contribution from all authors.

## Competing interests

The authors declare no competing interests.
