## [Peer Review File · Nature Communications]

REVIEWER COMMENTS

Reviewer #1 (Remarks to the Author):

In neo-Darwinian theory, adaptation results from a response to selection on relatively slowly accumulating genetic variation. A rapid path to adaptation may prove crucial when genetic variation is lacking. The Cichlid fishes of Lake Malawi display extensive phenotypic diversity despite a very low level of genetic divergence. This study aimed at generating high-coverage whole-genome bisulfite sequencing and total RNA sequencing of both liver and muscle tissues from 6 male, wild-caught Lake Malawi cichlid species to study the epigenetic basis for adaptive phenotypic diversification. The outcome of this extensive field-based study is timely and significant because the epigenetic basis for phenotypic variation in the field population was lacking. The paper is elegantly written. Figures and supplemental information provided are of high quality. A comparison of methylome profile with the liver and muscles' transcriptome is the strength as it was always difficult to connect observed epigenetic variation with the expression of genes in the same tissue.

The study is a descriptive epigenome-wide survey. The information it brings can be a basis for future research pertaining to epigenetic regulation of phenotypic plasticity in the natural species context. It is very well written; however, while reading through the manuscript, I could not make a solid conclusion. What would be the overall conclusion after integrating all those DMRs and transcriptional variations in all six species? Is “Our study suggests an epigenetic basis of adaptive phenotypic diversification in a spectacular example of rapid vertebrate radiation” the solid conclusion from such a comprehensive study? What is the brief message for the readers from a broad field of science and the public?

A significant critique is why authors used tissues to determine epigenetic variation as DNA methylation and histone modifications are highly variable within the homogeneous cell population. The central dogma of molecular biology states that, within a tissue containing multiple cell types, repressed genes utilize epigenetic marks of silencing. In contrast, the genes that express genes utilize permissive marks. What can be understood from sequencing the epigenome of a whole tissue? It is almost impossible to pinpoint what cell types within the tissues are responsible for the variations observed. This is a technical question that does not alter the outcome of the message this manuscript is trying to deliver; however, an effective approach is always necessary to solve biological questions in a convincingly scientific way.

Reviewer #2 (Remarks to the Author):

In the current study Vernaz et al describe base-resolution DNA methylomes of lake Malawi cichlids and propose that epigenetic variation played a key role in the rapid diversification of this species. The manuscript is well-written, easy to understand, and considerable effort has been invested in data presentation (and accompanying stats). The data generated for this study have an immense resource value and would likely on its own merit publication in Nature Communications. The manuscript, however, also has a major drawback; the current data do not support a role for 5mC in adaptive diversification. My criticisms are detailed below:

The authors perform relatively stringent mapping using Bismark (0-1 mismatches), however there is no mention of any sort of genotype correction being performed. Unless this step was included but not discussed in the manuscript, it is difficult to conclude if the identified DMRs are just a by-product of different genotypes (for example TG instead of CG). This can only be resolved by genotype correction, i.e. by looking at which nucleotide is found at which frequency on the opposite strand. Such a correction needs to be performed in its most stringent form and DMRs reanalyzed. Given the ambiguity of the bisulfite approach, it would also be important to display some examples of DMRs with their underlying BS-converted sequences as well as corresponding genomic sequences (from the same sample).

It is also important to keep in mind that a genetic change that influences TF binding can result in upstream or downstream changes in 5mC content (for example see Domcke et al, 2015; Nature). Also, if a TF is expressed in one tissue (or individual) but not the other, again perhaps due to a genetic change (i.e mutation, deletion etc), this can result in DMRs at or around the binding sites for that TF, between tissues (individuals). For example see Thurman et al, 2012; Nature (Fig.4). Authors could look at TF motif enrichment at DMRs and compare the expression of these TFs between samples / tissues. I strongly believe that all these issues need to be addressed if the authors really want to invoke an epigenetic basis for the observed cichlid diversity.

Finally, the manuscript offers very little support that any of these changes are adaptive. These conclusions are based on a number of anecdotal examples. To test this, the authors should design in vivo loss / gain of function assays where such adaptability could be measured.

I would suggest that the authors re-write the manuscript and change the title to something that would be more appropriate for a resource-type paper and discuss all the possibilities in a more balanced manner. Alternatively, much more work is needed to demonstrate that the cichlid diversity is driven by epigenetic (methylation in particular) and not genetic changes.

Reviewer #3 (Remarks to the Author):

Vernaz et al. performed a comprehensive study to test for a role of epigenetics in adaptive diversification in Lake Malawi cichlids. They performed whole-genome bisulfite sequencing and transcriptome sequencing to find associations between differentially methylated regions and differential gene expression. The authors found significant associations among species from different ecomorphological groups, relative to comparisons within species. They concluded that differences in methylation patterns might have contributed to adaptive diversification in this cichlid radiation.

I will defer to other reviewers to make specific comments on the validity of the methylome-transcriptome analyses. My expertise is in adaptive radiation and plasticity, not in whole-genome/methylome or transcriptome analysis. However, I can attest that this is a timely and important paper, and assuming that the methodology is sound, it is certainly worthy of publication in a prestigious journal such as Nature Communications. Its potential broad implications for understanding the molecular mechanism of biological diversity are of interest to a broad audience.

Specific issues to (easily) address:

-More transparency about sample sizes. Six species * 2 tissues is not entirely accurate. There were 6 species used in total, but only 3 for WGBS of liver, for example; and only 5 species for RNAseq. Sample sizes should be made apparent early on and continuously throughout the MS. Assume that readers will not refer to the supplementary information.

-Define acronyms at least during the first mention (e.g. line 61) and again in figures (e.g. GCI in Figure 1d) because some people only look at figures without reading the entire text. Defining acronyms will make the MS accessible to more people.

-How were fish euthanized? Was fish handling approved and conducted in accordance to animal use protocols?

-It's not possible to say that the DMR and differences in expression are adaptive without performing fitness studies. It is also not possible to determine that these differences drove diversification. These differences could have arisen due to random chance after diversification (since epigenes and expression patterns can be inherited, leading to similarities within species). Please acknowledge the limitations in inferring causality while shedding light on the evidence for the role of methylation in

adaptive diversification versus random processes. (Expand upon what is written on lines 398-399 – please highlight alternative explanations.)

The abstract is clear. The reference list is thorough.

Point-by-point rebuttal response to the reviewers; Revised manuscript Vernaz et al., NCOMMS-21-00548A

Reviewer #1 (Remarks to the Author):

In neo-Darwinian theory, adaptation results from a response to selection on relatively slowly accumulating genetic variation. A rapid path to adaptation may prove crucial when genetic variation is lacking. The Cichlid fishes of Lake Malawi display extensive phenotypic diversity despite a very low level of genetic divergence. This study aimed at generating high-coverage whole-genome bisulfite sequencing and total RNA sequencing of both liver and muscle tissues from 6 male, wild-caught Lake Malawi cichlid species to study the epigenetic basis for adaptive phenotypic diversification. The outcome of this extensive field-based study is timely and significant because the epigenetic basis for phenotypic variation in the field population was lacking. The paper is elegantly written. Figures and supplemental information provided are of high quality. A comparison of methylome profile with the liver and muscles' transcriptome is the strength as it was always difficult to connect observed epigenetic variation with the expression of genes in the same tissue.

- We thank the reviewer for their positive comments.

The study is a descriptive epigenome-wide survey. The information it brings can be a basis for future research pertaining to epigenetic regulation of phenotypic plasticity in the natural species context. It is very well written; however, while reading through the manuscript, I could not make a solid conclusion. What would be the overall conclusion after integrating all those DMRs and transcriptional variations in all six species? Is "Our study suggests an epigenetic basis of adaptive phenotypic diversification in a spectacular example of rapid vertebrate radiation" the solid conclusion from such a comprehensive study? What is the brief message for the readers from a broad field of science and the public?

- We thank the reviewer for its positive feedback. We have extensively improved the main text and the abstract to better convey the main messages and conclusions of this study.
- The main conclusion of our study is that we demonstrated substantial species-specific methylome divergence in closely related but phenotypically divergent cichlid species. Such divergence is significantly associated with transcriptional changes in ecologically-relevant genes among the livers of Lake Malawi cichlids and with altered activity of hepatic transcription factors, pointing to a link between dietary ecology and epigenetic divergence. Moreover, half of the methylome divergence is related to genes with functions related to developmental and embryonic processes. Altogether our work provides evidence for substantial species-specific epigenetic variation that might participate in phenotypic diversification associated with species-specific adaptations. It represents an important resource for further experimental work, aiming at assessing the adaptive advantage of such variation in the context of cichlid radiation, as well as the inheritance of such methylome patterns.

A significant critique is why authors used tissues to determine epigenetic variation as DNA methylation and histone modifications are highly variable within the homogeneous cell population. The central dogma of molecular biology states that, within a tissue containing multiple cell types, repressed genes utilize epigenetic marks of silencing. In contrast, the genes that express genes utilize permissive marks. What can be understood from sequencing the epigenome of a whole tissue? It is almost impossible to pinpoint what cell types within the tissues are responsible for the variations observed. This is a technical question that does not alter the outcome of the message this manuscript is trying to deliver; however, an effective approach is always necessary to solve biological questions in a convincingly scientific way.

- We thank the reviewer for pointing this out. We appreciate the limitations that arise from using bulk whole-tissue RNAseq (as opposed to single cell sequencing), but we chose this approach for two main reasons. First, whole tissue sequencing (RNAseq and WGBS) was done using two very homogenous tissues. Although any variation associated with cell subpopulations would not be examined here, whole tissue (bulk) RNAseq and WGBS provide a very valuable initial insight into the overall transcriptome and methylome variations in a whole tissue. Based on this work, our lab is now carrying out single-cell experiments to further elucidate the contribution of individual cell population variation (RNAseq, WGBS and ChipSeq among others). Second, RNA and DNA samples were extracted from wild-caught tissues over several field trips and were conserved in a highly saline solution (RNA/ater) to avoid DNA/RNA degradation - access to freezing devices were not possible during field trips. Such a preservation method is unfortunately not compatible with single-cell sequencing (cell lysis/permeabilization). We have now clearly highlighted the sequencing approach used in the method section and Suppl Figures.

Reviewer #2 (Remarks to the Author):

In the current study Vernaz et al describe base-resolution DNA methylomes of lake Malawi cichlids and propose that epigenetic variation played a key role in the rapid diversification of this species. The manuscript is well-written, easy to understand, and considerable effort has been invested in data presentation (and accompanying stats). The data generated for this study have an immense resource value and would likely on its own merit publication in Nature Communications. The manuscript, however, also has a major drawback; the current data do not support a role for 5mC in adaptive diversification. My criticisms are detailed below:

- We would like to thank the reviewer for their positive overall comment.

The authors perform relatively stringent mapping using Bismark (0-1 mismatches), however there is no mention of any sort of genotype correction being performed. Unless this step was included but not discussed in the manuscript, it is difficult to conclude if the identified DMRs are just a by-product of different genotypes (for example TG instead of CG). This can only be resolved by genotype correction, i.e. by looking at which nucleotide is found at which frequency on the opposite strand. Such a correction needs to be performed in its most stringent form and DMRs reanalyzed. Given the ambiguity of the bisulfite approach, it would also be important to display some examples of DMRs with their underlying BS-converted sequences as well as corresponding genomic sequences (from the same sample).

- We thank the reviewer for pointing this out and agree such a correction would make the analysis more accurate and avoid methylation biases due to genotype differences. As suggested by the reviewer, we have now generated SNP-corrected versions of the reference assembly *Maylandia zebra* (MZ) for each of the species/individuals analysed in this study. To this aim, we used high quality whole-genome sequencing data recently published by Malinsky et al (2018) and substituted high-confident individual-specific SNPs (C>T or G>A) to the MZ reference genome. The same method has been used in other methylome studies to account for different genotypes (see for example Schultz et al., Nature 2015 and Kawakatsu et al., Cell 2016). We chose this SNP-substituted genomes approach as it offers higher accuracy compared to the strategy consisting of using the other strand of WGBS reads to correct for genotype. This latter approach would require very high read coverage (~30) at CG sites for reliable SNP substitution (see for example Liu et al. 2012 Genome Biology), which would not be possible with our current dataset (8-15x genome coverage on average per sample). Methylation scores at CG sites across species are now accurate in all our analyses, reflect accurate methylation levels and do not result from SNP differences (see multi-alignment profiles of the SNP-corrected genomes in Appendix A, page 6).
- Using the respective SNP-corrected genomes, all methylome analyses were repeated for all samples. This included conservative mapping (0 mismatch allowed), CG methylation level extraction at CG sites only, and robust and large DMR calling (≥ 50 bp-long, 25% average mC difference, ≥ 4 CG sites). Overall mapping rates to SNP-corrected genomes were not significantly different ($57.5 \pm 2.9\%$ mapped paired-end reads; Dunn's corrected multiple testing, $P > 0.05$; Supp Fig. 2e).
- After comparison with the old dataset, this new method truly reflects methylation levels at CG sites, taking into account species-specific genotypes to avoid biases and improves DMR calling accuracy. On average, $84.7 \pm 3.6\%$ of all DMRs previously found between species/tissue pairwise comparisons could be identified again using the SNP-corr. genomes (see Table R1 below for comparison) and the total number of DMRs across pairwise comparisons remained similar overall (14.1+38 more DMRs compared to old DMR dataset). Approximately 15 % of all the DMRs were not predicted using the new dataset, due to genotyping corrections. We include two examples of DMRs used in the manuscript for which we show the methylome profiles before and after corrections (see Fig. R1, below) - both DMRs show robust hypomethylation in one species only compared to the others in both datasets, reflecting methylation levels at CG sites only conserved between species. We have also included the sequence alignment at the DMR for each of the species studied to illustrate sequence conservation and SNP substitution (see Fig R2 below) - we have not included this figure in the main manuscript as DMRs have now been called using SNP-corrected genomes that confidently reflect methylation levels (SNP-corrected genomes have been uploaded to GEO repository).
- Overall, the main results and conclusions of the paper remain unchanged after genotype correction, such as the proportion of multi-tissue and tissue-specific DMRs, and the identity of DMRs associated with changes in transcription (see revised figures and text).
- Throughout the manuscript, we have updated the main texts, figures, supp figures and materials/methods to take into account these new analyses and results (see edits and comments highlighted directly on the revised manuscript). Moreover, all new files (DMR list [bed] and methylome profile files [bwj]), as well as the SNP-corr. genomes (fasta) have been uploaded to GEO or have been added as supplementary materials.

A

Liver DMRs - between-species

Comparison	n_newDMR	n_oldDMR	n_overlap	p_overlap	n_diffCount
MxP	1,436	1,433	1,138	79.2%	3
AcxP	2,126	2,110	1,768	83.2%	16
AcxM	2,228	2,216	1,844	82.8%	12
AcxAs	2,349	2,338	1,980	84.3%	11
DxR	2,474	2,479	2,059	83.2%	-5
AsxP	2,569	2,585	2,164	84.2%	-16
AcxD	2,597	2,551	2,166	83.4%	46
AsxM	2,631	2,662	2,268	86.2%	-31
AcxR	2,663	2,616	2,199	82.6%	47
AsxD	2,728	2,734	2,318	85.0%	-6
PxD	2,866	2,900	2,429	84.8%	-34
DxM	3,059	2,992	2,512	82.1%	67
PxR	3,076	3,025	2,531	82.3%	51
AsxR	3,191	3,102	2,672	83.7%	89
RxM	3,225	3,263	2,687	83.3%	-38
Total (merged)	13,331	10,988			
Mean				83.4%	14.1
Sd				1.6%	38.5

B

Muscle DMRs - between-species

Comparison	n_newDMR	n_oldDMR	n_overlap	p_overlap	n_diffCount
PxR	2,436	2,394	2,051	84.2%	42
DxP	1,936	1,849	1,603	82.8%	87
DxR	1,765	1,709	1,464	82.9%	56
Total (merged)	4,390	4,236			
Mean				83.3%	61.7
Sd				0.8%	23.0

C

Tissue DMRs - within-species

Comparison	n_newDMR	n_oldDMR	n_overlap	p_overlap	n_diffCount
D	19,244	18,700	17,942	93.2%	544
R	11,400	10,969	10,524	92.3%	431
P	8,032	7,895	7,448	92.7%	137
Total (merged)	27,165	26,459			
Mean				92.8%	370.7
Sd				0.5%	210.1

Table R1. Tables showing the total number of DMRs predicted using SNP-corrected genomes ("newDMR"), including comparison with the old DMR dataset ("oldDMR"), for all liver (A), muscle (B) and tissue (C) pairwise comparisons. Absolute counts have the prefix "n_". Overlap (p_., percentage) representing the percentage of all the old DMRs found in the new DMR dataset. n_diffCount, represents the difference in total DMR number (newDMR - oldDMR). Sd, standard deviation.

Figure R1. Examples of methylome profiles at two genes containing a DMR, using SNP-corrected genomes for one multi-tissue embryonic DMR (a., from Fig.4e) and a putative functional DMR (pfDMR) associated with altered transcription in the livers (b., from Fig.3e). Methylome profiles using the old dataset (without genotype correction) are shown below ("pre-genotype corr."). All DMRs in the revised manuscripts have been predicted using SNP-corrected genomes.

It is also important to keep in mind that a genetic change that influences TF binding can result in upstream or downstream changes in 5mC content (for example see Domcke et al, 2015; Nature). Also, if a TF is expressed in one tissue (or individual) but not the other, again perhaps due to a genetic change (i.e mutation, deletion etc), this can result in DMRs at or around the binding sites for that TF, between tissues (individuals). For example see Thurman et al, 2012; Nature (Fig.4). Authors could look at TF motif enrichment at DMRs and compare the expression of these TFs between samples / tissues. I strongly believe that all these issues need to be addressed if the authors really want to invoke an epigenetic basis for the observed cichlid diversity.

- We thank the reviewer for raising this point. We have now included analysis for TF binding site enrichment associated with DMR sequence (see Supplementary Fig. 10d-f; text from line 402). We observe significant enrichment for some TF binding motifs in DMRs associated with differentially expressed genes among the livers of Lake Malawi cichlid fish. Interestingly, some of the TFs whose binding motifs are enriched are also differentially expressed in some species (such as the

transcription factor hepatocyte nuclear factor 4 alpha (hnf4a); Supp Fig.10d,e), suggesting a correlation between methylome divergence at binding sites and TF activity. Such TFs have functions related to hepatic gene expression regulation and lipid metabolism, possibly in line with adaptation to different diets.

- Importantly, we also observe enrichment for many binding sites bound by TFs with functions related to development and embryogenesis, such as foxk1 and foxa2 (Supp Fig. 10f; lines 484). Such TFs, not expressed in the livers, might reflect important changes in TF activity during development among the species analysed and may contribute to core phenotypic divergence established early on during development. We have discussed those results in the main text.
- It is known that many DNA-binding proteins, such as TFs, are methyl-sensitive, whose binding affinity might be affected by changes in methylation at their binding sites. Methylation changes at binding sites might also be associated with altered chromatin accessibility, therefore impeding TF binding affinity and leading to altered transcriptional network in a species-specific manner. Alternatively, as suggested by the reviewer, altered TF activity itself (due to SNP in the binding site or binding domain or due to the TF not being expressed) has been reported to manifest in methylation changes at TF binding sites as well. Both scenarios (cause/consequence - i.e., TF activity causing mC changes and TF activity altered following mC changes) have been reported in the literature (see Zhu, Wang, Qian, *Nat. Rev. Genet.* (2016)) and further work would also be required to elucidate whether any genetic basis might be correlated with TF activity and mC changes.
- We have now extensively discussed those results in detail (see Supp Fig10d-f) in the text (results and discussion; lines 661) as well as the crosstalk between TFs and DNA methylation. Relevant references have also been added.

Supplementary Fig. 10 (from manuscript). Selected panels from Supp Fig. 10 (from revised manuscript) on transcription factor (TF) binding motif enrichment analysis. **d.** Methylome divergence associated with altered transcription in livers reveals enrichment for specific binding motifs of differentially expressed TF genes in the livers (**e.**). Other TF binding motifs are enriched in multi-tissue DMRs and have functions related to liver development and embryogenesis.

Finally, the manuscript offers very little support that any of these changes are adaptive. These conclusions are based on a number of anecdotal examples. To test this, the authors should design in vivo loss / gain of function assays where such adaptability could be measured.

I would suggest that the authors re-write the manuscript and change the title to something that would be more appropriate for a resource-type paper and discuss all the possibilities in a more balanced manner. Alternatively, much more work is needed to demonstrate that the cichlid diversity is driven by epigenetic (methylation in particular) and not genetic changes.

- We agree with the reviewer and have extensively edited the manuscript and revised the title and abstract to better reflect the correlative/association nature of our study. Our work provides evidence for substantial methylome divergence in ecologically-relevant genes, with significant association with altered transcription network and altered transcription factor activity among closely related cichlid species. This study provides a rare and high-quality resource that paves way for further experimental work to further assess the extent to which these DMRs are adaptive, inherited and to understand any genetic basis of such epigenetic divergence.

References:

- T. Kawakatsu, S., et al., Epigenomic Diversity in a Global Collection of Arabidopsis thaliana Accessions. *Cell*. 166, 492–505 (2016). doi:10.1016/j.cell.2016.06.044
- Y. Liu, K. D. Siegmund, P. W. Laird, B. P. Berman, Bis-SNP: Combined DNA methylation and SNP calling for Bisulfite-seq data. *Genome Biol.* **13** (2012), doi:10.1186/gb-2012-13-7-r61.
- M. Malinsky, H. Svardal, et al., Whole-genome sequences of Malawi cichlids reveal multiple radiations interconnected by gene flow. *Nat. Ecol. Evol.* **2**, 1940–1955 (2018). doi:10.1038/s41559-018-0717-x
- H. Zhu, G. Wang, J. Qian, Transcription factors as readers and effectors of DNA methylation. *Nat. Rev. Genet.* **17**, 551–565 (2016). doi:10.1038/nrg.2016.83
- M. D. Schultz, Y. He, et al., Human body epigenome maps reveal noncanonical DNA methylation variation. *Nature*. **523**, 212–216 (2015). doi: 10.1038/nature14465

Reviewer #3 (Remarks to the Author):

Vernaz et al. performed a comprehensive study to test for a role of epigenetics in adaptive diversification in Lake Malawi cichlids. They performed whole-genome bisulfite sequencing and transcriptome sequencing to find associations between differentially methylated regions and differential gene expression. The authors found significant associations among species from different ecomorphological groups, relative to comparisons within species. They concluded that differences in methylation patterns might have contributed to adaptive diversification in this cichlid radiation.

I will defer to other reviewers to make specific comments on the validity of the methylome-transcriptome analyses. My expertise is in adaptive radiation and plasticity, not in whole-genome/methylome or transcriptome analysis. However, I can attest that this is a timely and important paper, and assuming that the methodology is sound, it is certainly worthy of publication in a prestigious journal such as Nature Communications. Its potential broad implications for understanding the molecular mechanism of biological diversity are of interest to a broad audience.

- We would like to thank the reviewer for their positive comments on the manuscript.

Specific issues to (easily) address:

- More transparency about sample sizes. Six species * 2 tissues is not entirely accurate. There were 6 species used in total, but only 3 for WGBS of liver, for example; and only 5 species for RNAseq. Sample sizes should be made apparent early on and continuously throughout the MS. Assume that readers will not refer to the supplementary information.

- We agree with the reviewer and have now revised the manuscript and the figures to clearly mention sample sizes for each analysis carried out (see Fig.1c, line 156). In total, 6 species were used for liver WGBS; 3 species for both liver and muscle WGBS; 4 species for both liver and muscle RNAseq.

-Define acronyms at least during the first mention (e.g. line 61) and again in figures (e.g. GCIs in Figure 1d) because some people only look at figures without reading the entire text. Defining acronyms will make the MS accessible to more people.

- We have defined all the acronyms throughout the text. We have also removed most abbreviations from the main figures to facilitate reading.

-How were fish euthanized? Was fish handling approved and conducted in accordance to animal use protocols?

- All the fish part of this study were bought dead from local fishermen under our supervision. Sampling collection and shipping were approved by permits issued to GF Turner, MJ Genner R Durbin, EA Miska by the Fisheries Research Unit of the Government of Malawi (various collaborative projects) and the Tanzania Fisheries Research Institute.
- We have added this information to the Methods section and to the Nature Communications Report Summary.

-It's not possible to say that the DMR and differences in expression are adaptive without performing fitness studies. It is also not possible to determine that these differences drove diversification. These differences could have arisen due to random chance after diversification (since epigenes and expression patterns can be inherited, leading to similarities within species). Please acknowledge the limitations in inferring causality while shedding light on the evidence for the role of methylation in adaptive diversification versus random processes. (Expand upon what is written on lines 398-399 – please highlight alternative explanations.)

- We agree that our study cannot conclude that the methylome divergence we observe among species is adaptive or drove phenotypic diversification. Further experimental work would be required to assess this issue. Our study rather highlights methylome divergence associated with both transcriptional changes in ecologically-relevant genes and altered transcription factor activity, pointing to a link between methylome divergence and dietary ecology. This study provides a rare and high-quality resource (methylome and transcriptome) in closely-related species and paves way for further experimental work aiming at assessing the heritability, plasticity (such variation could be purely plastic, in response to environmental context) and any genetic basis (any SNP in TF binding motifs, in TF binding domain sequence or in proteins involved in the maintenance and deposition of new methylC) underlying such natural epigenetic variation, as well as its potential adaptive role.
- Throughout the text, we have now extensively discussed and clearly stated the correlative nature of our work, the limitations associated with it, the other possible components underlying such epigenetic variation (alternative explanations), as well as the future experiments required to provide evidence for an adaptive role and heritability of such patterns. The abstract and the title have been changed to better reflect our results and conclusions as well.

The abstract is clear. The reference list is thorough.

- We thank the reviewer for this comment.

Appendix A1

Sequence alignment using species-specific SNP-corrected genomes at one pfDMR in the gene *tstd1*-like; see Fig.3e. CpG sites are numbered; DMR coordinates: NC_036781.1-27267631;27268527; 13 CpGs.

AC, *A. calliptera*; AS, *A. stuartgranti*; MZ, *M. zebra*; PG, *P. genalutea*; RL, *R. longiceps*; DL, *D. limnothrissa*

REVIEWERS' COMMENTS

Reviewer #2 (Remarks to the Author):

The authors did a great job answering the Reviewers' comments. I am happy to see that the overall results did not change much when the data was remapped to SNP-corrected reference genomes. Also, additional bioinformatics analyses have further strengthened the authors' conclusions. In my opinion the manuscript is now ready for publication.